# Chromenopyrimidinone Controls Stemness and Malignancy by suppressing CD133 Expression in Hepatocellular Carcinoma

**DOI:** 10.3390/cancers12051193

**Published:** 2020-05-08

**Authors:** Yeonhwa Song, Sanghwa Kim, Hyeryon Lee, Joo Hwan No, Hyung Chul Ryu, Jason Kim, Jee Woong Lim, Moonhee Kim, Inhee Choi, Haeng Ran Seo

**Affiliations:** 1Cancer Biology Laboratory, Institut Pasteur Korea, Seongnam-si 13488, Korea; yeonhwa.song@ip-korea.org (Y.S.); Sanghwa.kim@ip-korea.org (S.K.); 2Leishmania Research Laboratory, Institut Pasteur Korea, Seongnam-si 13488, Korea; hyeryon.lee@ip-korea.org (H.L.); celestial7@ip-korea.org (J.H.N.); 3R&D Center, J2H Biotech, Suwon-si 16648, Korea; daman-ryu@j2hbio.com (H.C.R.); jsbach@j2hbio.com (J.K.); jw0120@j2hbio.com (J.W.L.); mhkim@j2hbio.com (M.K.); 4Medicinal Chemistry, Institut Pasteur Korea, Seongnam-si 13488, Korea

**Keywords:** liver cancer stem cells (LCSCs), CD133, hepatocellular carcinoma (HCC), UBE2C, chromenopyrimidinone (CPO)

## Abstract

Hepatocellular carcinoma (HCC) is a highly malignant human cancer that has increasing mortality rates worldwide. Because CD133^+^ cells control tumor maintenance and progression, compounds that target CD133^+^ cancer cells could be effective in combating HCC. We found that the administration of chromenopyrimidinone (CPO) significantly decreased spheroid formation and the number of CD133^+^ cells in mixed HCC cell populations. CPO not only significantly inhibited cell proliferation in HCC cells exhibiting different CD133 expression levels, but also effectively induced apoptosis and increased the expression of LC3-II in HCC cells. CPO also exhibits in vivo therapeutic efficiency in HCC. Specifically, CPO suppressed the expression of CD133 by altering the subcellular localization of CD133 from the membrane to lysosomes in CD133^+^ HCC cells. Moreover, CPO treatment induced point mutations in the ADRB1, APOB, EGR2, and UBE2C genes and inhibited the expression of these proteins in HCC and the expression of UBE2C is particularly controlled by CD133 expression among those four proteins in HCC. Our results suggested that CPO may suppress stemness and malignancies in vivo and in vitro by decreasing CD133 and UBE2C expression in CD133^+^ HCC. Our study provides evidence that CPO could act as a novel therapeutic agent for the effective treatment of CD133^+^ HCC.

## 1. Introduction

Hepatocellular carcinoma (HCC) is the seventh most common malignant tumor and the third leading cause of mortality worldwide [1,2,3,4]. Notably, HCCs are mainly resistant to common chemotherapeutic agents, and HCC patients typically have poor tolerance of systemic chemotherapy due to underlying liver dysfunction. Consequently, there is no standard therapy for HCC at present. Further, the cumulative 3-year relapse rate after liver resection is about 80%, and relapse after operation is associated with a high mortality rate [5]. Hence, a highly efficacious drug that targets the liver and induces fewer side effects would fill up a critical need in liver cancer drug discovery. Many researches have sought to determine drugs candidates and target genes for HCC, but the development of a therapeutic drug for liver cancer has not yet been successful.

Cancer stem cells (CSCs) were determined after experiments in which tumor cells were fractionated and characterized based on cell-surface markers and injected into mice at limiting dilutions. The cell populations that induced the tumor growth that subsequently led to tumor growth following transplantation of the initial tumor into a second animal were classified as CSCs [6]. Owing to their strong resistance to radiotherapy and chemotherapy, CSCs are considered an ‘Achilles heel,’ following to tumor metastasis, recurrence, and treatment failure [7,8,9]. Hence, recent advances using HCC stem cells to develop effective anti-tumor agent have been promptly recognized as a new goal.

In the previous study, we identified and fractionated CSCs in primary HCC patients and determined that CD133 was a cell-surface marker of liver cancer stem cells (LCSCs) [10]. CD133^+^ cells presented increased tumor spheroid capacity, stem-like properties, chemoresistance, the ability of migration and tumorigenic capacity compared to CD133^−^ cells in HCC. From a clinical perspective, HCC patients with increased CD133 expression have poor overall survival and increased recurrence rates compared to liver cancer patients with low CD133 expression [11,12]. Current research have likewise indicated that CD133 is a novel target for overcoming chemoresistance in HCC [13]. To overcome CD133-induced chemoresistance, the functional role of CD133 in chemoresistance must be elucidated and new targets or compounds that target CD133^+^ cells need to be identified. Accordingly, we performed an image-based high content screening (HCS) to identify compounds that particularly target CD133^+^ HCC cells present in mixed populations of hepatocyte and HCC cells. Consequently, we determined chromenopyrimidinone (CPO; ChemDiv C201-0053), which selectively inhibited CD133^+^ HCC cells. In addition, D133 promoted autophagy and was translocated from the membrane to the cytoplasm [14]. This may be needed for CSC survival in the tumor microenvironment [15,16].

In this study, we found that CPO was associated with the induction of apoptosis and autophagy in HCC. We also found that CPO reduced autophagy levels in CD133^+^ HCC cells. Likewise, CPO altered the subcellular localization and induced the degradation of CD133 in HCC cells. As a result, CPO suppressed the stemness and malignancy of CD133^+^ HCC cells. These results suggest that combined treatment with CPO and conventional anticancer therapies could serve as a novel clinical strategy against liver cancer that specifically targets CSC-like cells, preventing damage to normal tissue cells.

## 2. Results

### 2.1. CPO Effectively Suppresses CD133^+^ Cells in HCC

In a previous study, we identified CSCs from primary HCC and determined that CD133 is a CSC cell-surface marker [10,17,18]. Additionally, we conducted screening to identify the compounds that particularly altered the properties of AFP^+^/CD133^+^ cells as LCSC using a well-defined mixed HCC cell population and HCS imaging technology in Fa2N-4 cells (immortalized hepatocyte line) and Huh7.5 cells [18]. In the current research, we sought to identify new compounds targeting both HCC (AFP^+^/CD133^−^) and LCSC (AFP^+^/CD133^+^) versus normal hepatocytes (AFP^−^/CD133^−^) in mixed HCC cell populations. After screening, we identified the compound, chromenopyrimidinone (CPO), which inhibited the activity of the AFP^+^/CD133^+^ cell population and the proliferation of AFP^+^/CD133^−^ cells without damaging hepatocytes in mixed HCC cell populations (Figure 1A).

To find previously reported biological assays related to the CPO compound, we searched the PubChem Bioassay database (Figure 1B) (National Center for Biotechnology Information. PubChemDatabase, CID = 135572401, https://pubchem.ncbi.nlm.nih.gov/compound/135572401 (accessed on Feb. 19, 2020)). Our search returned a total of nine biological assays for CPO, all of which were for various viruses and bacteria. It was concluded to be inactive in an inhibition assay of CDC25B-CDK2/CyclinA interaction. In addition, we searched the ChEMBL database [19], but the search returned no reported biological assays. Hence, we concluded that there were no reported assays for CPO related to cancer. To determine the inhibitory effects of CPO on AFP^+^/CD133^−^ and AFP^+^/CD133^+^ cells, the dose-response of CPO was measured in mixed HCC cell populations. Remarkably, CPO showed more sensitive effects in AFP^+^/CD133^-^ cells (IC_50_ 35.0 nM) and AFP^+^/CD133^+^ cells (IC_50_ 37.9 nM) than in AFP^−^/CD133^−^ cells (IC_50_ 344.4 nM) (Figure 1C).

Because CSCs are abundant in non-adherent spheroids of liver, colon, and breast cancer cells, we sought to determine whether CPO alters the malignant properties of CSC populations in HCC. We treated 200 nM CPO, 10 nM taxol, 10 µM cisplatin, and 10 µM sorafenib under Huh7 spheroid-forming conditions and analyzed the number of spheroids formed. Notably, CPO sufficiently attenuated the capacity of CD133^+^ HCC to form spheroids compared to taxol, cisplatin, and sorafenib (Figure 1D). To determine the effect of CPO on CD133^+^ HCC cells, we picked four human HCC lines that display different expression levels of CD133 in the following order: Huh7 > Hep3B > PLC/PRF/5 > Huh6 (Figure 1E). Interestingly, when these HCC cell lines were treated with CPO, the IC_50_ value for CPO was inversely proportional to CD133 expression in the Huh6 (1.3 µM) > PLC/PRF/5 (1.2 µM) > Huh7 (413.8 nM) > Hep3B (464.8 nM) cells (Figure 1F). In addition, a dose-response curve also presented that the cell death increased by CPO in HCC cells (Huh7, Hep3B), which contain an abundant population of CD133^+^ cells compared to normal hepatocytes (Fa2N-4) (Figure 1G). Notably, immunohistochemistry revealed that CPO selectively attached to the AFP^+^/CD133^+^ HCC cells in a co-culture system of hepatocyte and HCC cells (Figure 1H).

### 2.2. CPO Induces Apoptosis in HCC Cells

To confirm whether the CPO-induced inhibition of cell growth was related to an increase in apoptosis, we conducted a western blot assay and looked at the apoptosis-related parameters though V-FITC/PI flow cytometry. We observed the early and late apoptotic phases with treatment of indicated concentrations of CPO in both cells including Huh7 and Hep3B. Significant dose-dependent increases (*p* < 0.01) in the number of apoptotic cells following CPO treatment were only observed in Huh7 and Hep3B cells, and not Fa2N-4 cells (Figure 2A).

To determine why CPO had different effects on cell death in hepatocytes versus HCC cells, we investigated the stability of the CPO compound in media from hepatocytes and HCC cells following cell culture. We found that the residual amounts of CPO were lower in media from HCC cells compared to media from hepatocytes (Figure 2B).

We used the Sub-G1 assay to measure apoptosis, which is widely used as it is easy, rapid, reliable, reproducible, and economical. Using flow cytometry, we estimated the Sub-G1 peak in CPO-treated Huh7 cells and Hep3B cells and then found that treatment with 0.4 µM CPO induced an increase in the Sub-G1 population in Huh7 cells in a time-dependent manner (Figure 2C). To investigate the specific apoptotic mechanisms induced by CPO, we confirmed cleaved-poly (ADP-ribose) polymerase (PARP) and cleaved caspase-3 expression levels by western blot analysis in Hep3B cells and Huh7 cells, respectively. Caspase-3, a member of the interleukin-1 beta-converting enzyme family, is involved in the induction of apoptosis. As such, cleaved caspase-3 and cleaved PARP are considered as markers for measuring apoptosis, an active form of caspase-3 and PARP. The cleaved caspase-3 and cleaved PARP expression levels were increased in CPO-treated Hep3B and Huh7 cells in a dose-dependent manner (Figure 2D).

3D cell culture systems accurately recapitulated the 3D cellular context and therapeutically related pathophysiological gradients of the tumor in vivo, including pH and oxygen gradients, growth factor penetration rates, and the distribution of proliferating/necrotic cells. Hence, we investigated the effects of CPO on HCC cell line-derived spheroids and found that the size of HCC cell line-derived spheroids was significantly reduced by CPO treatment (Figure 2E).

### 2.3. CPO Induces Autophagy in HCC Cells

Because the CD133^+^ HCC populations with hypoxic or malnutrition conditions are increased to the level of autophagy [18] and CPO effectively suppresses CD133^+^ HCC, we observed effects of CPO on autophagy parameters.

To investigate if the reduction in cell growth induced by CPO treatment was related to autophagy mechanisms, we performed diverse autophagy parameters by the CYTO-ID Autophagy Detection kit, as well as immunohistochemistry and western blotting. We detected the number of spots representing the potential autophagosome formation in the cytoplasm of cells using the autophagy detection kit. The number of spots dramatically increased in the positive control group (5 µM rapamycin, autophagy inducer), but decreased in the negative control group (50 µM chloroquine, autophagy inhibitor). Interestingly, the number and area of the spots significantly increased with increasing concentrations of CPO (Figure 3A,B). Autophagy—a type II programed cell death—is a complicated, multi-step process executed following intracellular membrane/vesicle reconstruction for forming double-membraned autophagosomes, which ultimately degrade their contents by acidic lysosomal hydrolases. Microtubule-associated protein 1 LC3 was specifically used as an autophagy detection marker that is converted from LC3-I (inactivation form) to LC3-II (activation form) during autophagy. We found that the expression level of LC3-II significantly increased in a dose-dependent manner in CPO-treated Huh7 and Hep3B cells (Figure 3C). Immunohistochemistry also revealed that CPO increased LC3 puncta to induce autophagy in HCC cells (Figure 3D). These results indicated that autophagy is related to CPO-induced cytotoxicity in HCC.

### 2.4. CPO Altered CD133 Subcellular Localization and Induced Degradation of CD133 in HCC

Because we determined that CPO was specific for LCSCs (AFP^+^/CD133^+^ cells), we then performed studies on the function of CPO in CD133^+^ cells. Our previous study reported that the degradation of CD133 appears to be via the endosomal-lysosomal pathway [20]. In accord with these results, we observed CD133 internalization in Huh7 cells following treatment with CPO. We traced CD133 in Huh7 cells by immunohistochemistry using lysosomes labelled with LysoTracker. An immunofluorescence-based analysis showed that with 0.05% DMSO, the expression of CD133 on the lysosome and cell membrane labelled using LysoTracker appeared in the cytoplasm; however, with treatment of CPO in Huh7 cells, which presented the loss of cell-surface CD133, resulted in the appearance of higher levels of CD133 with LysoTracker in the cytoplasm (Figure 4A).

Because the internalization of CD133 degrades performance, we observed changes in total CD133 expression following CPO treatment. We conducted a western blot assay to confirm CD133 expression after CPO treatment in Hep3B and Huh7 cells. Interestingly, the treatment of CPO caused reductions in the CD133 expression level (Figure 4B) and mRNA (Figure 4C) in a dose-dependent manner. Because CPO released CD133 from the plasma membrane to the cytoplasm, we detected CD133 expression at the membrane and in the cytoplasm after treatment with CPO. However, CPO treatment decreased CD133 expression at the membrane and in the cytoplasm in Huh7 and Hep3B cells (Figure 4D). In a previous study, we confirmed that CD133 expression was associated with the metastatic ability of HCC cells in vitro [10]. Because CPO inhibited CD133 expression, we estimated the metastatic capacity in Huh7 and Hep3B cells in the current study using a cell migration assay, and found that CPO suppressed metastatic capacity in Huh7 and Hep3B cells in a dose-dependent manner (Figure 4E).

### 2.5. RNA-seq Profiling Reveals a Novel Target Pathway of CPO in HCC

As a regulator of CSCs, CD133 has diverse functional roles including tumor growth, autophagy, chemoresistance, and lipid metabolism [18]. Our results showed that CPO decreased CD133 expression; thus, we analyzed the effects of CPO-induced CD133 degradation on genome-wide alterations in gene expression in HCC using RNA-seq, an influential approach for investigating drug-induced changes in genome-wide gene expression. Determining the mechanisms of drug action in human cells is still an important challenge. To identify signaling pathways and genes downstream of CPO, we performed subsequent pathway analysis and RNA sequencing analysis. Using expression levels ≥2 as an arbitrary cutoff [9] for authentic gene expression, we detected 198 dose-dependent Differentially Expression Genes (DEGs). Among them, 123 genes exhibited increased expression, whereas 75 were decreased. RNA-seq revealed that CPO mainly affected cell cycle, metabolism, and post-transcriptional protein modification (Figure 5).

We sought to identify the potential targets of CPO based on the RNA-seq data as well as the structural characteristics of the compound. The sequencing data showed that the genes modulated by CPO are largely related to cell cycle. Among them, some genes were reported to indirectly interact with topoisomerase I or modulated by its inhibitor camptothecin [21,22,23]. In a structural perspective, CPO is similar to camptothecin in harboring four planar rings leading to possible inhibition of topoisomerase I by CPO. To evaluate this possibility, molecular docking simulation was performed. The docking was first validated with self-docking of camptothecin to topoisomerase I (PDB: 1T8I) showing a high degree of alignment between the X-ray structure and predicted pose (Figure 6A).

In the docking with CPO, the predicted pose aligns well to camptothecin with the score of −9.367 compared to −10.156 for camptothecin. The prediction was validated by a DNA relaxation assay. As shown in Figure 6B, CPO completely inhibited the relaxation activity of the enzyme at 10 µM, shown by the presence of unrelaxed supercoiled DNA at the bottom of the gel, and by the dose-response curve, where the EC_50_ value was calculated as 3.6 μM. Even with some extent of inhibition, the values were relatively higher than the IC_50_ value on the growth of HCC (Figure 1), implying topoisomerase I may not be the primary target, but possibly involved in the control of stemness and malignancy of HCC.

In particular, treatment with CPO induced point mutations in four of the genes including adrenoceptor beta 1 (*ADRB1* G389R), apolipoprotein B (*APOB* P2739L), early growth response protein 2 (*EGR2* K396E), and ubiquitin conjugating enzyme E2C (*UBE2C* S23R) (Table 1).

*EGR2* is related to adipocyte differentiation and cholesterol metabolism [24,25], whereas mutations in and abnormal levels of *APOB* affect cholesterol levels [26,27]. *ADRB1* is involved in the lipolysis of stored triglycerides in adipocytes [28,29]. *UBE2C*, an exclusive partner of APC/C [30], is overexpressed in HCC and facilitates the malignant potential of HCC cells [31].

### 2.6. CPO Target Cell Cycle and Lipid Metabolism Pathway in HCC

Next, we examined whether point mutations in the four genes affected the expression of ADRB1, APOB, EGR2, or UBE2C in Huh7, Hep3B, and SNU449 cells and found that the expression of all four proteins were thoroughly reduced by CPO treatment (Figure 6C).

In order to elucidate the mechanism by which CPO compounds can efficiently kill CD133 cells unlike other anticancer drugs, we searched for proteins that were affected by CD133 expression among the four proteins. We observed the effects of CD133 deletion on the expression of the four proteins using siRNA against CD133 in Huh7 and Hep3B cells. We confirmed that the expression of UBE2C only decreased among the four proteins by siRNA for CD133 in both cell lines (Figure 6D).

Because UBE2C is essential for the degradation of mitotic cyclins, cell cycle distribution was examined after the treatment of CPO in HCC. The G2/M phase were increased after 12 hours of treatment of CPO in Hep3B cells (+6.1%) and Huh7 cells (+12.3%) (Figure 6E).

Our results found that CPO restricted the proliferation of HCC cells by inhibiting UBE2C, and controlled lipid metabolism-related proteins such as ADRB1, APOB, and EGR2.

### 2.7. CPO Exhibits In Vivo Therapeutic Efficiency in HCC

In a previous study, we demonstrated that CD133^+^ cells exhibited higher and faster tumor growth and tumorigenecity compared to CD133^−^ cells, suggesting that CD133^+^ cells can provoke HCC tumor formation. As the results, we injected Huh7 CD133^+^ cells orthotopically into NOD/SCID mice to analyze LCSC suppression by CPO. Because CPO is a partially soluble drug, it can be injected intra-peritoneal (I.P.) in mice. Despite poor solubility, the administration of 5 mg/kg CPO inhibited tumor growth without a loss in body weight compared to mice treated with 10 mg/kg sorafenib (data not shown), which resulted in only a slight reduction in tumor growth (Figure 7A,B).

These results suggest that CPO treatment suppresses the LCSC properties and CD133^+^ HCC cell population by destabilizing CD133 in CD133^+^ HCC cells; thereby CPO facilitates in vivo therapeutic efficiency in HCC.

## 3. Discussion

An ultimate goal in liver cancer research is to discover drug candidates that selectively target HCC CSCs. CD133^+^ HCC cells may, however, be the “Achilles heel” of HCC. Therefore, we developed a state-of-the-art, mixed HCC cell population using HCC cells, liver cancer stem cells (LCSC) and hepatocytes, in accord with a published mixed culture system [32]. In a previous study, we identified CPO following a non-target-based high-throughput screening to identify compounds to eliminate LCSCs (AFP^+^/CD133^+^cells) and HCCs (AFP^+^/CD133^−^ cells) without inducing toxicity in hepatocytes (AFP^−^/CD133^+^ cells) in mixed populations of HCC cells with hepatocytes. The results of a proliferation assay revealed that CPO significantly suppressed the proliferation of CD133^−^ and CD133^+^ HCC cells in a dose-dependent manner (Figure 1C,F), and other analyses demonstrated that CPO induced autophagy and apoptosis in HCC cells (Figure 2 and Figure 3). Apoptosis is a major mechanism by which cytotoxic agents induce tumor cell death. We suggest that CPO induced apoptosis results in inducing apoptosis related proteins such as cleaved caspase-3 and cleaved PARP. Similar to the effects observed in vitro, CPO also exhibited efficient therapeutic efficiency in HCC in vivo (Figure 5).

Targeting autophagy is promising for cancer therapy, but it has been also controversial due to its multifaceted roles in regulating cell survival death. In our study, we demonstrated that CPO treatment induces autophagy in HCC cells. We determined that an abundance of autophagic vacuoles and the expression of the autophagy-related LC3-II protein increased by CPO induced autophagy (Figure 3). It is important to note that autophagy plays an important role in anti-cancer therapy tumorigenesis; otherwise, many previous studies that investigated autophagy in different cell types and under different conditions have yielded conflicting results. Recent studies suggested that some of the signaling pathways, including those mediated by mTOR, AKT, ROS, phosphoinositide 3-kinase (PI3K), and HIF-1, have been involved in the regulation of autophagy [19,20,21]. Further investigation is needed to determine whether these signaling pathways were implicated in autophagy induced by CPO in HCC cells.

In this study, we suggested that CPO induced autophagy and apoptosis in HCC cells and LCSCs. A recent study which demonstrated the relationship between autophagy and apoptosis and the crosstalk between the apoptosis marker, Bcl-2, and autophagy marker, Beclin-1, has attracted increasing attention. This study presented that the interaction with Bcl-2 and Beclin-1 inhibited autophagy [33]. They suggested that further studies are needed to investigate the relationship between apoptosis and autophagy for cancer therapy.

After identifying CPO, we determined the potential mechanisms of action in LCSC. To ascertain the effects of CPO on CD133^+^ HCC cells, we estimated the CD133^+^ HCC cell population in HCC cells with abundant CD133^+^ HCC cell populations. Interestingly, CPO treatment decreased the expression of CD133 by altering the subcellular localization of CD133 (Figure 4A,B). The recent study demonstrated the interaction between CD133 and HDAC6, histone deacetylase, and determined the mechanism of CD133 internalization and trafficking into lysosomes [20]. Although a similar function of receptor tyrosine kinases—another type of surface molecule that is internalized into the cell nucleus—has been reported [24,25,26,27,28,29,30,31,32,33,34,35,36], CD133 is a distinct class of cell membrane proteins, and the analogies to this process are therefore limited [37].

Although some studies have demonstrated that reductionist approaches such as target-based screening is useful for drug discovery, phenotypic screening is receiving renewed attention, because it may overcome the limitations for new findings [38]. However, targets or target of molecules identified through phenotypic screening have often been found to be impossible or slow. Consequently, it is important to identify potential drug target genes in drug discovery [39,40]. Herein, we utilized the RNA-sequencing approach, to determine the global transcriptional effects of CPO on HCC in order to accelerate the process of drug target identification (Figure 5) [41].

After performing RNA-seq, we concluded that UBE2C is a CPO target and is essential to induce cell death in HCC cells (Figure 6). UBE2C exerts oncogenic effects on various human solid cancers including breast cancer [42], HCC [31], colon cancer [43], nasopharyngeal carcinoma (NPC) [44] and, among others. UBE2C is essential for the degradation of mitotic cyclins, which subsequently enables UBE2C to accelerate cell proliferation and malignant transformation [45]. According to an analysis of the GEO [46] and TCGA (https://www.cancer.gov/tcga) databases, patients with HCC that exhibited increased UBE2C expression had a significantly shorter overall survival. Because inhibiting UBE2C expression can elevate radiation and chemosensitivity [47] and increase apoptosis, UBE2C is considered a potential therapeutic target for cancer therapy [48,49].

We determined that CPO inhibits UBE2C by inducing the degradation of UBE2C, indicating that CPO could be a potential HCC therapy. However, treatment with CPO could simultaneously induce a point mutation in UBE2C at S23R; thus, further functional studies of the UBE2C mutation at S23R are needed in HCC.

In addition, treatment with CPO reduced the expression of and induced point mutations in ADRB1 (G389R), APOB (P2739L), and EGR2 (K396E) (Table 1), which are all involved in lipid metabolism as well as cholesterol and triglycerides. Among them, ADRB1 R389G has been reported to be associated with chronic diseases related to obesity such as high blood pressure, atherosclerosis, and fatty liver [50,51,52]. According to our results, we expect that CPO will have potential applications not only to HCC but also to various metabolic diseases caused by obesity because CPO affected the point mutation at the same position in the studied Huh7 cells. Furthermore, EGR2 and abnormal apolipoprotein B levels are related to adipocyte differentiation and cholesterol metabolism. Thus, we need to further investigate whether CPO-induced CD133 degradation affects EGR2 and apolipoprotein B expression levels, to overcome nonalcoholic fatty liver disease.

## 4. Materials and Methods

### 4.1. Cell Lines and Culture Conditions

The Hep3B, Huh7, PLC/PRF/5, and SNU449 HCC cell lines were purchased from the Korean Cell Line Bank (Seoul, Korea). Huh7.5 cells were kindly provided by Dr. Marc Windisch (Institut Pasteur Korea, Gyeonggi-do, Korea), and Huh6 cells were obtained from Cell Bank Australia (Westmead, NSW, Australia). Human immortalized hepatocytes (Fa2N-4) were obtained from XenoTech (Lenexa, KS, USA). All cells were maintained at 37 °C in a humidified atmosphere of 5% CO_2_. Dulbecco’s Modified Eagle’s Medium (Welgene, Daegu, Korea) was used to cultivate the Hep3B, Huh7.5, and Huh6 HCC cell lines, and Roswell Park Memorial Institute 1640 (RPMI1640) medium was used to cultivate the Huh7, PLC/PRF/5, and SNU449 HCC cell lines. DMEM and RPMI1640 media were supplemented with heat inactivated 10% fetal bovine serum (FBS; Gibco, Grand Island, NY, USA) and 1× penicillin-streptomycin (P/S; Gibco). Fa2N-4 cells were plated on collagen-coated plates (BD Biosciences, San Jose, CA, USA) in serum-containing plating medium (plating media; XenoTech), which was replaced with supporting culture medium (XenoTech) after cell attachment (approximately 3–6 h).

### 4.2. Mixed Culture System

Fa2N-4 and Huh7.5 cells were seeded together in DMEM at densities of 1.2 × 10^3^ cells and 0.8 × 10^3^ cells, respectively, in 384-well culture plates (Greiner Bio-One, Monroe, NC, USA). After overnight incubation, the cells were treated with CPO for 48 h and stained with CD133 and AFP antibodies to distinguish normal hepatocytes (AFP^−^/CD133^−^), cancer cells (AFP^+^/CD133^−^), and cancer stem cells (AFP^+^/CD133^+^).

### 4.3. Drug Sensitivity of Liver Cancer Stem Cell Spheroids

Huh7 cells were seeded in low-attachment 6-well plates (Corning, NY, USA) at a density of 3 × 10^3^ cells/well with or without 10 nM Taxol, 10 µM Cisplatin, or 0.2 µM CPO for 7 days. The stem cell permission media was composed of DMEM/F12 (Gibco) supplemented with 1× B27 (Invitrogen, Eugene, OR, USA), 20 ng/mL basic fibroblast growth factor (bFGF; Invitrogen), 20 ng/mL epidermal growth factor (EGF; Invitrogen), and 25 µg/mL insulin (Sigma-Aldrich, St. Louis, MO, USA). After incubation, the spheroids were observed using Operetta HCS system (Perkin Elmer, Waltham, MA, USA).

### 4.4. Western Blot Analysis

The cells were suspended using a lysis buffer (Biosesang, Gyeonggi-do, Korea) and the resulting samples were boiled for 10 min in 5× sample buffer (Biosesang). After sample preparation, equal amounts of protein (10–30 µg/well) were separated on 8 or 10% SDS-PAGE gels. After electrophoresis, the proteins were transferred onto a nitrocellulose (NC) membrane (Pall Corporation, Port Washington, NY, USA) and blocked with 5% skim milk for 30 min at room temperature (R.T.). After blocking, the NC membranes were incubated with mouse monoclonal anti-CD133 (W6B3C1; 1:200; Miltenyi Biotec, Bergisch Gladbach, Germany), mouse monoclonal anti-β-actin (1:1,000; Sigma-Aldrich), rabbit monoclonal anti-cleaved PARP (Asp214) (1:1000; Cell Signaling Technology, Danvers, MA, USA), rabbit monoclonal anti-cleaved caspase-3 (Asp175) (1:1000; Cell Signaling Technology), rabbit polyclonal anti-LC3A/B (1:1000; Cell Signaling Technology), rabbit polyclonal anti-AKT (1:1000; Cell Signaling Technology), rabbit polyclonal anti-NA^+^/K^+^-ATPase (1:1000; Cell Signaling Technology), mouse monoclonal anti-apolipoprotein B (ApoB; 1:1,000; Novus Biologicals, Centennial, CO, USA), rabbit polyclonal anti-β-1 adrenergic receptor (ADRB1; 1:1000; Invitrogen), rabbit polyclonal anti-UBE2C (1:1000; Cell Signaling Technology), or rabbit polyclonal anti-EGR2 antibodies (1:1000; Novus Biologicals) for 16 h at 4 °C. After washing three times with Dulbecco’s Phosphate-Buffered Saline (DPBS; Welgene), the membranes were incubated with horseradish peroxidase-conjugated secondary antibody (Cell Signaling Technology) at a 1:10,000 dilution, and the specific bands were visualized by enhanced chemiluminescence (ECL; Thermo Scientific, Waltham, MA, USA).

### 4.5. High Content Screening (HCS) Imaging Assay Technology

Normal hepatocyte and HCC cells were plated in a 384-well plate at a density of 2.5 × 10^3^ cells/well and incubated for 72 h. After treatment with the indicated concentration of CPO, the cells were washed with DPBS and fixed in 4% paraformaldehyde (PFA; Biosesang) for subsequent fluorescent probe or antibody staining. Automated multispectral image acquisition was performed on the Operetta HCS system using a 20× objective. The fluorescence images were acquired according to the optimal excitation and emission wavelengths of each probe and secondary antibody. To capture a sufficient number of cells (>100) for analysis, five image fields were collected from each well, starting at the center of the well. Image analysis was performed using the Harmony software (Perkin Elmer).

### 4.6. Immunocytochemistry

Cells were fixed in 4% PFA for 10 min, and incubated with mouse monoclonal anti-CD133 (AC133/1, 1:100, Miltenyi Biotec) or rabbit polyclonal anti-AFP (1:200, Dako, Denmark A/S, Denmark) in DPBS with 10% normal goat serum (Vector Laboratories, Burlingame, CA, USA) and 0.01% Triton X-100 (Sigma-Aldrich) for 16 h at 4 °C. After washing with DPBS three times, the cells were incubated with a fluorescence-conjugated secondary antibody (Invitrogen) at a 1:500 dilution in DPBS for 1 h at R.T. in the dark. After washing with DPBS three times, the nuclei were stained with Hoechst 33342 (Invitrogen) for 10 min. Images were obtained using an Operetta HCS system. For lysosome staining, the cells were incubated with LysoTracker Red DND-99 (Invitrogen) in media for 20 min. The cells were then washed twice with DPBS and gently fixed in 4% PFA to avoid cell detachment.

### 4.7. Apoptosis Assay

The FITC Annexin V Apoptosis Detection Kit I (BD Biosciences, 556547) was used to detect apoptosis. Fa2N-4, Hep3B, and Huh7 cells were treated with or without 200 nM or 400 nM CPO for 24 h and the cells and supernatant were collected. Apoptosis was measured according to the manufacturer’s instructions.

### 4.8. Cell Cycle Analysis

The cells and supernatant were collected after treatment with 200 nM CPO for 6, 12, or 24 h. After washing with DPBS, the cells were resuspended in 100 µL of DPBS followed by the addition of 5 µL of PI. After incubation for 10 min, the cells were washed with DPBS and analyzed by flow cytometry.

### 4.9. Drug Sensitivity in HCC Spheroids

Huh7 and Hep3B cells were seeded in 80 µL of media at a density of 6 × 10^3^ cells/well in low-attachment 96-well plates (7007; Corning) and incubated for 3 days. The cells were then treated with 20 µL of 5× CPO at the 0.1, 0.2, 0.3, 0.5, or 0.8 µM and the spheroids were incubated for 4 days. Images were obtained using an Operetta HCS system with a 10× objective.

### 4.10. Autophagy Analysis

To measure autophagy, HCC cells including Huh7 and Hep3B were replaced in 384-well culture plates and treated with 0.1, 0.2, 0.3, or 0.4 µM of CPO for 48 h. Autophagy was detected using the CYTO-ID Autophagy Detection kit (ENZ-51031, Enzo Life Sciences, Lausen, Switzerland) following to the manufacturer’s instructions. Rapamycin (R) and chloroquine (CQ) were used as positive controls. The images were obtained using an Operetta HCS system, and the areas of the spots detected were analyzed using Harmony software.

### 4.11. Cell Fractionation

To determine the CD133 levels after CPO treatment, Hep3B and Huh7 cells were treated with 200 nM or 400 nM CPO for 48 h and the cells were collected for fractionation. To separate the cytosol and membrane fractions from the cell pellet, a subcellular protein fractionation kit (Thermo Fisher Scientific, P178840) was used following the manufacturer’s instructions. NA^+^/K^+^-ATPase was used as a membrane marker, and AKT was used as the cytosol marker.

### 4.12. Cell Migration Assay

Cell migration following treatment with 400 nM CPO for 24 or 48 h was evaluated using a Radius 384-well cell migration assay (CBA-127, Cell Biolabs Inc., San Diego, CA, USA) according to the manufacturer’s instructions. Briefly, 25 µL of Radius gel pretreatment solution was added to the Radius 384-well cell migration plate, followed by incubation at R.T. for 20 min. After aspirating the solution from each well, the wells were washed with Radius wash solution. Huh7 and Hep3B cells were seeded at a density of 3 × 10^3^ cells/well and 25 µL of 3× Radius gel removal solution was added and the plates were incubated in 37 °C incubator for 30 min. The Radius gel removal solution was removed by aspiration, and media and 400 nM CPO were added. After incubation for 24 or 48 h, images of cell migration were obtained using the Operetta HCS system.

### 4.13. Drug Efficacy in HCC Orthotopic Mouse Model

To generate the HCC orthotopic mouse model, HepG2-luciferase cells were used. A total of 1 × 10^6^ cells/30 µL of serum-free RPMI1640 media were injected into the left lobe of the liver in 6-week old female NOD/SCID mice (Central Lab. Animal Inc., Seoul, Korea). After 3 weeks, the bioluminescence (BLI) of the tumor was measured by an optical imaging system (IVIS spectrum CT, Perkin Elmer), and the mice were grouped following the results of the imaging system. The experiment groups are listed below.

Group 1: Vehicle control (10% DMSO + 90% Captisol)Group 2: 5 mg/kg CPOGroup 3: 10 mg/kg CPOGroup 4: 10 m/kg Sorafenib (positive control)

Intraperitoneal injection (I.P.) of CPO, sorafenib, and vehicle was performed daily for 2 weeks (5 times/week). The BLI of the tumor was measured three times a week by IVIS-spectrum CT after injection of 15 mg/kg luciferin (100 µL/mice) (GoldBio technology, St Louis, MO, USA) in the right and left abdomen of the mice. The body weight was measured at the same day of imaging. On the day of sacrifice (1 week later from finishing the compounds injection), the mice were anesthetized with 1 mL/kg of zoletil (Virbac Korea, Seoul, Korea) and rompun (Bayer Korea, Seoul, Korea) mixture with 2:1. Imaging of liver tissue was performed at the sacrifice day. All animal experiments was approved by the institutional animal care and use committee of ASAN medical center and conducted strictly in accordance to the national institute of health guide for the care and use of laboratory animals (ethic code: 2019-02-009).

### 4.14. Preparation for RNA Sequencing and Analysis

The selection of DEGs was based on the RNAseq results using CLRNASeq^TM^ software (http://www.chunlab.com/software_clrnaseq_download; Chunlab, Seoul, Korea). The data were normalized using the following methods: reads per kilobase per million mapped reads (RPKM), relative log expression (RLE), and trimmed mean of M-value (TMM). Genes that were up- and down- regulated more than two fold were selected based on DESeq and EdgeR values (*p*-value < 0.005); Single-Nucleotide Variant (SNV), and indel variant calling of selected genes were translated to respective protein sequences in order to analyze amino acid changes affected by CPO treatment.

### 4.15. Molecular Docking Simulation

For the docking simulation, the *Homo sapiens* topoisomerase I protein (PDB: 1TBI) was prepared using the protein preparation module in Maestro 11.0 (Maestro, Schrödinger, LLC, New York, NY, USA, 2019). Water molecules as well as the camptothecin ligand were removed from the active site. Hydrogen bonds were optimized using default value and an energy minimization was performed only on the protein hydrogens in Macro-Model (MacroModel, Schrödinger, LLC, New York, NY, USA, 2019). A large enough grid encompassing the active site was generated and the ligands prepared with protonation state at pH7.4 were docked using Glide (Glide, Schrödinger, LLC, New York, NY, USA, 2019) extra-precision mode without any constraints.

### 4.16. Topoisomerase I DNA Relaxation Assay

Inhibition of *Homo sapiens* topoisomerase I activity by CPO was evaluated by plasmid relaxation assay. Serially diluted CPO from 100 μM to 0.1 nM was incubated with the enzyme (PROSPEC, Ness-Ziona, Israel) for 2 hours and the reaction was carried on with the presence of supercoiled pBR322 plasmid DNA (Thermo Fisher Scientific Inc., Waltham, MA, USA) for 0.5 hours at 37 °C. The reaction was stopped by sodium dodecyl sulfate solution plus proteinase K, and agarose gel electrophoresis was performed to visualize the bands. The inhibition was quantified by measuring the intensity of supercoiled DNA band followed by calculation of EC_50_ values in Prism7.

### 4.17. Statistical Analysis

All experiments were conducted at least three times. The results are presented as the mean ± standard deviation (SD). A Student’s *t*-test was used to assess statistically significant differences in excel. The significances were considered respectively with *p* values of *p* < 0.05, *p* < 0.01, and *p* < 0.001.

## 5. Conclusions

In conclusion, our findings indicate that CPO-induced cell death and degradation of UBE2C and CD133 by CPO in HCC cells are associated with both the apoptotic and autophagic pathways. Moreover, our data presented that CPO may be a promising therapy for HCC, enabling patients to overcome chemoresistance and tumor recurrence due to CSCs in HCC. As well, CPO may have therapeutic potential for treating lipid-related metabolic diseases. Nonetheless, the function of ADRB1, APOB, EGR2, and UBE2C and a more detailed understanding of the molecular mechanisms underlying the effects of CPO in HCC cells needs further investigation. Further prospective clinical cancer trials will be required to determine whether CPO will prove to be a useful and safe anticancer agent.

## Figures and Tables

**Figure 1 cancers-12-01193-f001:**
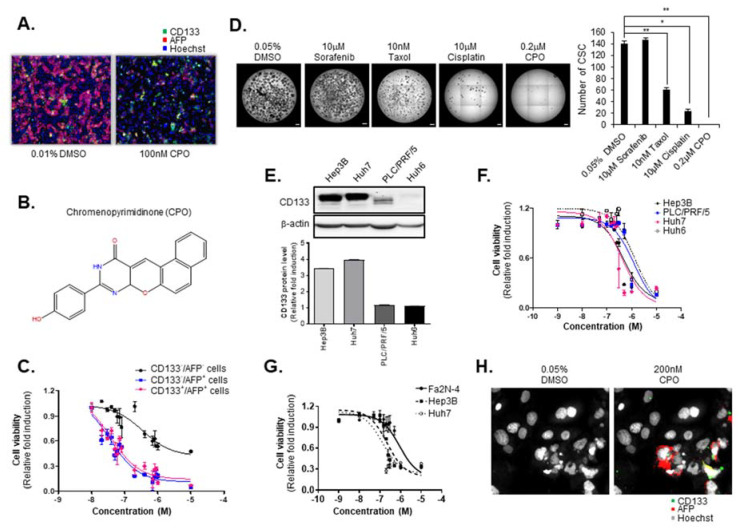
Suppression of liver cancer stem cells (LCSCs) and hepatocellular carcinoma (HCC) by chromenopyrimidinone (CPO). (**A**) Normal hepatocytes, cancer cells, and CSCs in a mixed culture system using Fa2N-4 and Huh7.5 after treatment with 100 nM CPO for 48 h. (**B**) Chemical structure of chromenopyrimidinone (CPO). (**C**) Viability of normal hepatocytes (AFP^−^/CD133^−^), cancer cells (AFP^+^/CD133^−^), and CSCs (AFP^+^/CD133^+^) in mixed culture systems at the indicated concentration. (**D**) CSC spheroid formation after treatment with 10 nM Taxol, 10 µM cisplatin, 10 µM sorafenib and 0.2 µM CPO for 3 days (left panel). The number CSC were counted (right panel). Scale bar = 500 µm. (**E**) Expression of CD133 in the Hep3B, Huh7, PLC/PRF/5, and Huh6 hepatocellular carcinoma cell lines (upper panel). Expression of CD133 was quantified (lower panel). β-actin was used as the control. (whole blot image, Appendix A). (**F**) Viability of HCC cell lines (Hep3B, PLC/PRF/5, Huh7, and Huh6) after treatment with the indicated concentration of CPO for 48 h. (**G**) Viability of Fa2N-4, Hep3B, and Huh7 cells after treatment with the indicated concentration of CPO for 48 h. (**H**) Morphology of cancer cell and CSC nuclei after treatment with 200 nM CPO for 48 h. Images were acquired using a HCS system with a 40× objective. The data shown are from three independent experiments relative to the control values and the mean values ± SD from three independent experiments; * *p* < 0.05 and ** *p* < 0.01 compared to CPO treatment group.

**Figure 2 cancers-12-01193-f002:**
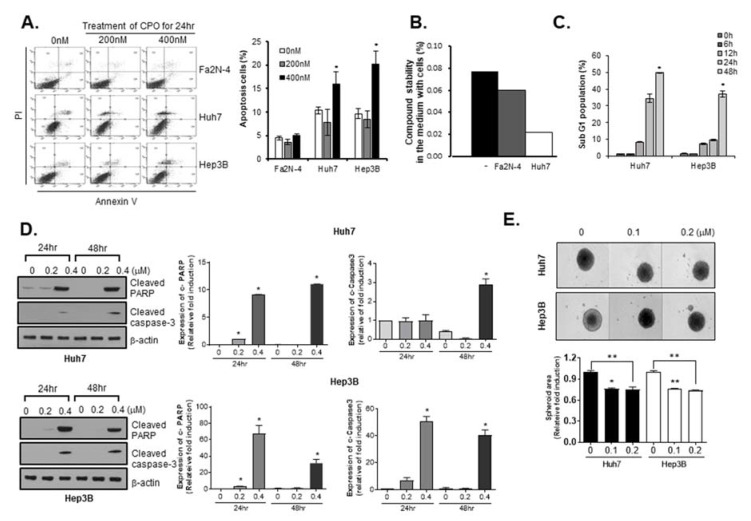
Apoptosis in hepatocellular carcinoma (HCC) induced by chromenopyrimidinone (CPO). (**A**) Annexin V/PI positive cells (apoptotic cells) in Fa2N-4, Huh7, and Hep3B cells after treatment with 200 nM or 400 nM CPO for 24 h determined by flow cytometry (left panel). Graph of percentages of apoptotic cells (right panel) detected by flow cytometry. * *p* < 0.05 compared to untreated group. (**B**) Percentages of CPO stability in the media from Fa2N-4 and Huh7 cells. * *p* < 0.05 compared to control group. (**C**) Percentages of cell cycle phase (SubG1) after treatment with 200 nM CPO for 6, 12, 24, or 48 h determined by flow cytometry. Graph of cell phase percentages determined by flow cytometry. (**D**) Expression of apoptosis-related proteins (cleaved PARP, cleaved caspase-3) after treatment with or without 200 nM or 400 nM CPO for 24 h or 48 h in Huh7 (upper panel) and Hep3B (lower panel) cells. Expression of protein was quantified (right panel). The whole blot image can be found in Appendix A. (**E**) Size of Huh7 and Hep3B spheroids after treatment with the indicated concentration of CPO for 4 days. Spheroid area was quantified (bottom panel). Images were obtained using an HCS system. Scale bar = 500 µm. The data shown are from three independent experiments relative to the control values and the mean values ± SD from three independent experiments; * *p* < 0.05 and ** *p* < 0.01 compared to control group.

**Figure 3 cancers-12-01193-f003:**
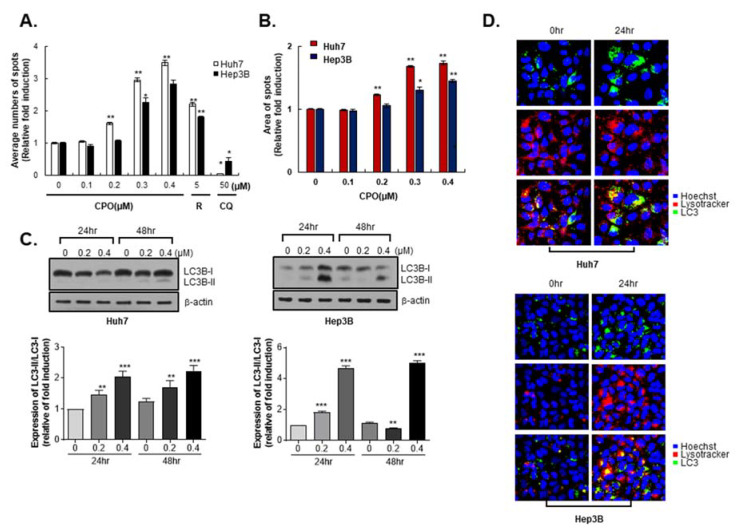
Autophagy in hepatocellular carcinoma (HCC) induced by chromenopyrimidinone (CPO). (**A**) Average numbers of autophagosome spots after treating Huh7 and Hep3B cells with the indicated concentration of CPO for 48 h. Rapamycin (R) was used to induce autophagy, and chloroquine (CQ) was used to inhibit autophagy. (**B**) The area of autophagosome spots after treating Huh7 and Hep3B cells with the indicated concentration of CPO for 48 h. (**C**) Expression of LC3B-I and LC3B-II after treat Huh7 (left panel) and Hep3B cells (right panel) with or without 200 nM or 400 nM CPO for 24 h or 48 h (upper panel). Expression of proteins was quantified (lower panel). β-actin was used as the control (whole blot image, Appendix A). (**D**) Immunocytochemistry images of LysoTracker (red) for detecting lysosomes, LC3 (green), and the nucleus (blue) after treating Huh7 (upper panel) and Hep3B cells (lower panel) with 200 nM CPO for 24 h. Images were acquired using an HCS system with a 40× objective. The data shown are from three independent experiments relative to the control values and the mean values ± SD from three independent experiments; * *p* < 0.05, ** *p* < 0.01 and *** *p* < 0.001 compared to control group.

**Figure 4 cancers-12-01193-f004:**
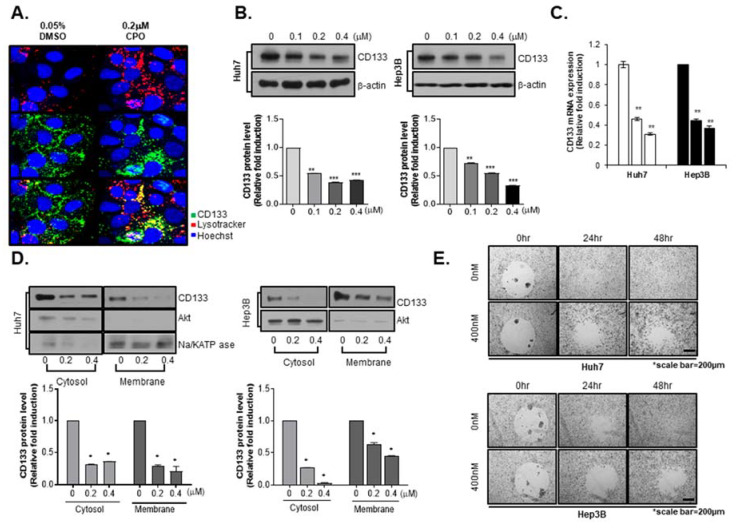
Subcellular localization and degradation of CD133 induced by chromenopyrimidinone (CPO). (**A**) Immunocytochemistry images of CD133 (green), lysosomes (red), and the nucleus (blue) after treating Huh7 cells with or without 200 nM CPO for 48 h. Images were obtained using an HCS system with a 40× objective. (**B**) Expression level of CD133 after treating Huh7 (left panel) and Hep3B (right panel) cells with 100 nM, 200 nM, or 400 nM CPO for 48 h. Expression of proteins was quantified (lower panel). β-actin was used as the control (whole blot image, Appendix A). (**C**) Expression level of CD133 mRNA after treating Huh7 and Hep3B cells with 200 nM or 400 nM CPO for 48 h. β-actin was used as the control. (**D**) CD133 expression levels in the cytosol and membranes after treating Huh7 (left panel) and Hep3B (right panel) cells with or without 200 nM or 400 nM CPO for 48 h. Expression of CD133 protein was quantified (lower panel). NA^+^/K^+^-ATPase was used as a membrane marker, and AKT was used as a cytosol marker. The whole blot image can be found in Appendix A. (**E**) Cell migration capacity after treating Huh7 and Hep3B cells with 400 nM CPO for 24 h or 48 h. Scale bar = 200 µm. All results shown are from three independent experiments. Data are expressed as mean ± SD; * *p* < 0.05, ** *p* < 0.01 and *** *p* < 0.001 compared to control group.

**Figure 5 cancers-12-01193-f005:**
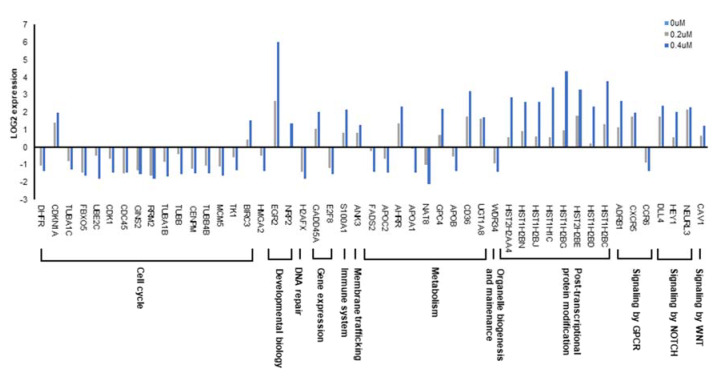
RNA-seqeuncing (RNA-seq) profiling reveals a novel target pathway of CPO in HCC. Genome-wide expression of the indicated signal pathway-related genes in Huh7 cells treated with 200 nM or 400 nM CPO determined by RNA-sequencing analysis.

**Figure 6 cancers-12-01193-f006:**
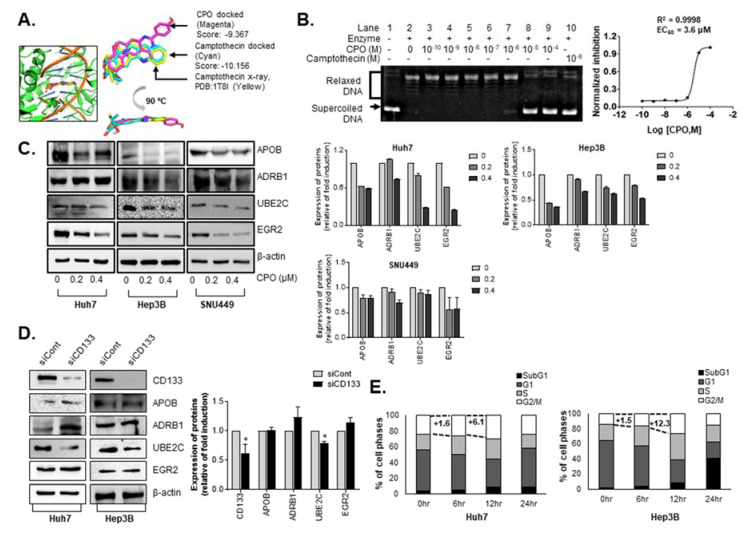
CPO inhibits expression of ADRB1, APOB, EGR2, and UBE2C in HCC. (**A**) Predicted binding mode of CPO (in magenta) located between the nicked DNA bases in topoisomerase I active site with a superimposition to X-ray structure (in yellow) and docked pose (in cyan) of camptothecin. (**B**) Inhibition of human topoisomerase I activity by CPO evaluated using DNA relaxation assay in dose-diluted manner. (**C**) RNA-sequencing results of the expression of novel target proteins in Huh7, Hep3B, and SNU449 cells after treatment with 200 nM or 400 nM CPO for 48 h (left panel). Expression level of proteins was quantified (right panel). The whole blot image can be found in Appendix A. (**D**) Expression of novel target protein in CD133-depelted HCC cell lines (Huh7 and Hep3B) (left panel). Expression of proteins in siCont and siCD133 was quantified (right panel). β-actin was used as the control (whole blot image, Appendix A). (**E**) Percentages of cell cycle phase (SubG1, G1, S, and G2/M) after treatment with 200 nM CPO for 6, 12, and 24 h determined by flow cytometry in Hep3B and Huh7 cells (left panel). Graph of percentages of cell phase was shown (right panel). The data shown are from three independent experiments relative to the control values and the mean values ± SD from three independent experiments; * *p* < 0.05 compared to control group.

**Figure 7 cancers-12-01193-f007:**
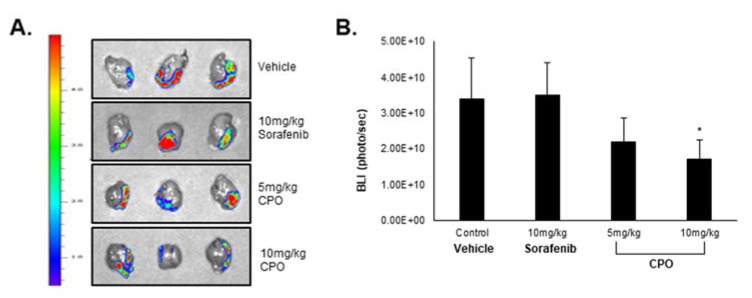
Efficacy of chromenopyrimidinone (CPO) in an orthotopic hepatocellular carcinoma mouse model. (**A**) HCC tumor size in the mouse model after being treated with 5 mg/kg and 10 mg/kg of CPO for 2 weeks. The fluorescence in the tumor images is bioluminescence (BLI) through optical imaging system. (**B**) The intensity of BLI in each group. Data are expressed as mean ± SD; * *p* < 0.05 compared to control group.

**Table 1 cancers-12-01193-t001:** CPO induced point mutations in Huh7 cells.

Gene Name	Full Name	Position	Ref.	Alt.	Quality *	Mutation
ADRB1	Adrenoceptor beta 1	114045297	G	C	149.9	G389R
APOB	Apolipoprotein B	21008652	G	A	9889.7	P2739L
EGR2	Early growth response 2	62813452	T	C	31.77	K396E
		62813453	C	T	31.77	K396E
UBE2C	Ubiquitin-conjugating enzyme E2C	45812764	C	G	813.77	S23R

* Quality score of respective SNV and indel variant calling (≥20).

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
