# Peer review of "Chromenopyrimidinone Controls Stemness and Malignancy by suppressing CD133 Expression in Hepatocellular Carcinoma"

_cancers, 2020, doi:10.3390/cancers12051193_

Round 1

Reviewer 1 Report

Song et al. proposed a novel drug candidate for HCC treatment, Chromenopyrimidinone, which seems to regulate CD133. Although their proposal is interesting, there seems to be several, critical limitations.

・Chromenopyrimidinone was not clearly mentioned in the previous study (ref#18). Please explain more details of the screening method used and how they identified this compound.

・Why they used Taxol and cisplatin in Figure 1 experiments. These two drugs are not common for HCC treatment. How was the result of sorafenib, lenvatinib, or regorafenib?

・CPO did not clearly affect the autophagy status in Huh7 cells.

・From the RNA seq analysis, four genes such as ADRB1, APOB, EGR2, and UBE2C were affected with point mutations. However, their association with CD133 is unclear. Please show the mechanistic insights into the association, not only the results from siCD133 experiment.

・In Figure7, they show only BLI. How about CD133 expression, autophagy status, ADRB1, APOB, EGR2, and UBE2C expression? In addition, liver function and liver histology need to be analyzed using vivo samples.

Others:
・References #1-4 are too old to mention the current situation of HCC.

・There are grammar errors including basic ones.

Author Response

Thank you for your kind comments. 

we have uploaded the response to the reviewer's comments file.

Reviewer 2 Report

Well written paper of major interest for those working in this field of research.

No major or minor comments from my side.

Author Response

(The authors gave the same response as above.)

Reviewer 3 Report

The manuscript by Song et al., investigates the anti-cancer agent of chromenopyrimidinone (CPO) against hepatocellular carcinoma (HCC). More specifically the manuscript focuses on CD133, which is an important marker of cancer stem cells. The manuscript is interesting and detailed in proving the efficacy of CPO in suppressing CD133 and also in proving the mechanism of CPO activity. The efforts taken by the authors to present this research study is appreciated. 

  1. Figure 1 - The effects of CPO are compared to taxol and cisplatin. While the Western blotting data is convincing for the observed effects on CD133, the immunofluorescence picture (H) is not convincing and the authors need to provide better resolution and better stained pictures. 
  2. Figure 2 - A-C is missing statistical evaluations. Include the same.
  3. Figure 4 - How confident the authors are with the CD133 staining? Please provide details of the antibody used in the manuscript and cross verify the staining specificity. For example in DMSO treated cells, the staining pattern is completely different from CPO treated cells. It is imperative that the stainings are specific and I recommend the authors to be extra cautious in presenting the immunofluorescence staining.
  4. Dosage used - The initial dose used was 200 nM and the following experiments also included 400 nM dose. Please include the rational for the same in the manuscript. 
  5. RNA Seq profiling, molecular docking and DNA relaxation assay - The efforts taken by the authors to present these experiments are appreciated.
  6. Use of sorafenib - The authors needs to provide rational for the use of sorafenib as a control in the in vivo experiments. 
  7. Other comments - Please check for grammatical errors and few of the sentences requires modifications. Please check the manuscript for language edits during resubmission. 

Author Response

(The authors gave the same response as above.)

Reviewer 4 Report

The study is addressed to characterize a  novel compoud, namely chromenopyrimidinone (CPO), for the control of cancer stem cells (CD133+) in hepatocellular carcinoma.

The approach could be sound and mixed HCC cell populations in co-colture with immortalized hepatocytes have been carried out to study  different sensitivity and cell modifications induced by CPO. In addition, the effect of CPO has also assessed in experiments using NOD/SCID mice.

The main problem of the present paper is that it is very difficult to understand since several sentences are not intellegible. In addition, the desctiption of the results is unclear and the reader cannot perceive the details.

The interpretation of data is also puzzling, since it has been described that CPO alters CD133 subcellular localization and induces its degrdation  (see paragraph 2.4 of the Results), while in  paragraph 2.6 mechanism of CD133+ cell killing is considered.

It has been reported that CPO determines  reduction of Huh7 and Hep3B spheroid formation in a dose-dependent manner, however in figure 2E no dose-dependence reduction of the spheroid size can be observed, since the difference is significant only in relation to “0”, which  should mean absence of CPO.

Reported results on metabolism are weak. In Figure 6D the efficiency of CD133 siRNA is low and significance of  the extent of UBE2C decrease is not reported.

Data obtained in NOD/SCID mice are  not convincing and additional results, including kinetics of weight loss and tumor volume should be provided.

The Discussion should be improved.

Author Response

(The authors gave the same response as above.)

Round 2

Reviewer 1 Report

All the experiments were performed well.

In the discussion section, authors added the explanation of the previous study.

I think the MS has been well-improved.

There are some typo, please check.

Reviewer 4 Report

The authors have addressed the criticisms previously itemized and the manuscript has been improved.

This manuscript is a resubmission of an earlier submission. The following is a list of the peer review reports and author responses from that submission.